# An Attribute-Based Access Control Model in RFID Systems Based on Blockchain Decentralized Applications for Healthcare Environments

**Santiago Figueroa [1,2,*]** , **Javier Añorga [1,2]** and **Saioa Arrizabalaga [1,2]**

1   Ceit, Manuel Lardizabal 15, 20018 Donostia/San Sebastián, Spain
2   Departamento de Ingeniería Eléctrica y Electrónica, Universidad de Navarra, Tecnun, Manuel Lardizabal 13, 20018 Donostia/San Sebastián, Spain.
*   Correspondence: sfigueroa@ceit.es; Tel.: +34-943-213076

**Abstract:** The growing adoption of Radio-frequency Identification (RFID) systems, particularly in the healthcare field, demonstrates that RFID is a positive asset for healthcare institutions. RFID offers the ability to save organizations time and costs by enabling data of traceability, identification, communication, temperature and location in real time for both people and resources. However, the RFID systems challenges are financial, technical, organizational and above all privacy and security. For this reason, recent works focus on attribute-based access control (ABAC) schemes. Currently, ABAC are based on mostly centralized models, which in environments such as the supply chain can present problems of scalability, synchronization and trust between the parties. In this manuscript, we implement an ABAC model in RFID systems based on a decentralized model such as blockchain. Common criteria for the selection of the appropriate blockchain are detailed. Our access control policies are executed through the decentralized application (DApp), which interfaces with the blockchain through the smart contract. Smart contracts and blockchain technology, on the one hand, solve current centralized systems issues as well as being flexible infrastructures that represent the relationship of trust and support essential in the ABAC model in order to provide the security of RFID systems. Our system has been designed for a supply chain environment with an use case suitable for healthcare systems, so that assets such as surgical instruments containing an associated RFID tag can only access to specific areas. Our system is deployed in both a local and Testnet environment in order to stablish a deep comparison and determining the technical feasibility.

**Keywords:** blockchain; smart contract; RFID; ABAC; access control; IoT security; healthcare security; healthcare RFID system

## 1. Introduction

The healthcare field is aware of the essential need to adopt and use healthcare information technology (IT) successfully. Radio-frequency Identification (RFID) provides several opportunities for healthcare transformation [1]. The same reference before argues that RFID provides an enhanced method to decrease errors in patient-care, to improve tracking and tracing for both patients and equipment, as well as to enable better management of health assets and improving the audit process and predictability.

In general terms, four sub-systems describe an RFID system's architecture (Figure 1): (1) a transponder or tag, which contains the identification data, (2) a reader to interact directly with the tag exchanging information with it, (3) a RFID middleware and (4) a business and/or information management layer. RFID middleware supports RFID tag data management by handling devices,

filtering, collecting, integrating and constructing data. The business layer also includes applications such as back-end databases (DBs), enterprise resource planning (ERP), customer relationship management (CRM), warehouse management solutions (WMS), tracking and tracing and electronic product code (EPC) applications.

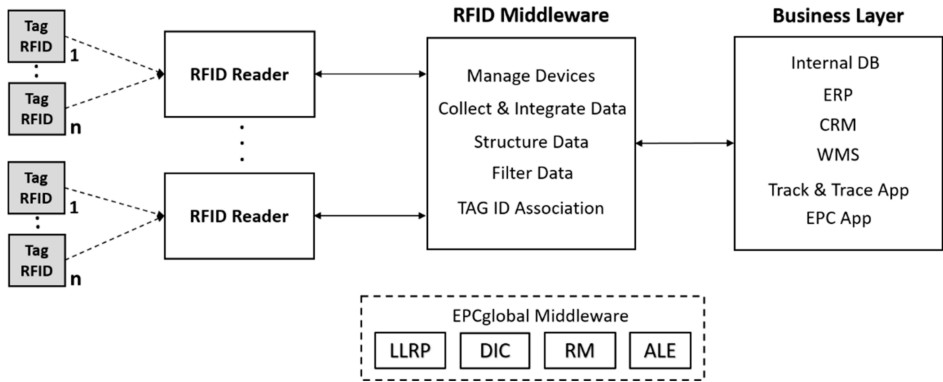

**Figure 1.** General architecture of Radio-frequency Identification (RFID) systems.

The GS1 (Whilst "GS1" is not an acronym it refers to the organization offering one global system of standards) standards ([2]) address three wide categories: identify, capture and share [3,4]. The capture process could be performed through sub-systems (1) and (2), using EPC-enabled for RFID tags (i.e., through EPC, GS1 also provides a construction to write and read unique identifiers on RFID tags). The identification process would be covered by sub-system (3) and an identification numbers is performed, for instance, when it is encoded (e.g., to GTIN (Global Trade Item Number)) or it is decoded (e.g., from RFID Tag EPC). Finally, the sub-system 4) carried out the sharing category. In particular, GTIN describes a data structure that uses 14 digits with the option to encode in some combinations. GTIN is currently used in both barcodes and RFID [2]. The structure of the GTIN number is shown below:

$$\text{urn:epc:id:sgtin:CompanyPrefix.ItemReference.SerialNumber,} \tag{1}$$

These fundamental principles are used to explain how the GS1 standards system can be used to enable traceability solutions, where RFID systems are involved in both data capture and data sharing. In addition, RFID systems are able to achieve traceability in a variety of supply chains such as fresh food, health, technical industries, transportation and logistics. The supply chain in the healthcare sector will be taken as use case. In that sense, RFID is the industry-leading technology used by medical device manufacturers to enable smart devices to provide higher-quality patient care, the most common RFID applications include [5]:

(1)   Tracking and tracing of trust device to individual patients.
(2)   Ensuring appropriate sterilization.
(3)   Control of servicing and calibration of medical equipment.
(4)   Invoicing procedures, to associate patients with medical device and prescription use.
(5)   Stock management.
(6)   Decreased time spent by staff tracking articles and devices.

In this way, a model based on control and traceability of assets is a determining factor in safety. Based on the analysis of the six points mentioned above, interviews with specialists were carried out in order to determine the needs existing in institutions, where a specific use case related to point 1) has been identified. Hospitals employ large numbers of assets (e.g., surgery medical instruments (SMI)), which can flow through constant cycles such as sterilization department, surgery room, laboratories, etc. A location mistake could risk the patients' lives. In addition, the lack of detailed asset records causes asset losses.

However, given that RFID is one of the most well positioned technologies to perform the data capture and sharing process, the biggest challenge for any RFID systems is its security. The security threats encountered in RFID systems are distinct from traditional wireless security threats, which can be grouped into: (1) physical components of RFID (e.g., cloning tags, reverse engineering, tag modification), (2) the communication channel (e.g., eavesdropping, skimming, replay attack), and 3) global system threats (e.g., spoofing, Denegation of Service (DoS) and "tracing and tracking"). Other examples and details can be obtained from reference [6]. Therefore, our proposal must focus on both safety risks and security risks.

Access control (AC) is a core piece of any organization's security infrastructure. In particular, AC has popularized as a solution for some of the threats mentioned [7]. Below is an overview of our proposal. Based on GS1, SMI are tagged (passive RFID tag) with GTIN. The coding scheme (see expression (2)) contains a company prefix (e.g., Hospital A: 000389), an article reference (product type) to categorize the asset (e.g., scissors: 000162) and, finally, a serial number to identify a specific asset (e.g., serial number: 000169740). Figure 2 helps to detail how our healthcare system works. The source room (e.g., sterilization department) sends some assets (e.g., SMI) to the destinations rooms (e.g., surgery room$_0$, surgery room$_1$). Since asset$_1$ has been assigned to destination room$_1$ (e.g., surgery room$_1$) and due to human mistake (e.g., in transportation) attempts to access to destination room$_0$ (e.g., surgery room$_0$), our system establishes access denied status to asset$_1$.

In short, our proposal is an access control system of a healthcare asset (e.g., SMI) in order to prevent unwanted assets from entering the wrong area (e.g., room) because of human error or an external security threat. Therefore, our proposal is a prevention system that provides a security-safety solution.

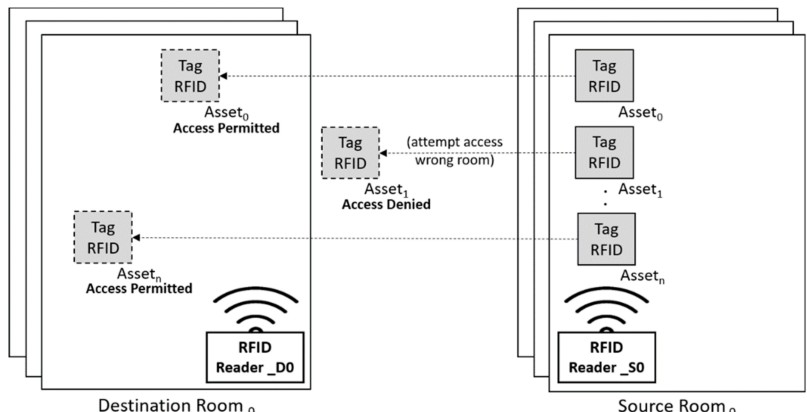

**Figure 2.** Healthcare system.

By considering the general model presented in Figure 1, RFID middleware commonly deploys the access control mechanism (ACM) in an RFID system [8]. For instance, the EPC global community standardizes four main layers within the middleware (Figure 1): low level read protocol (LLRP), discovery initialization and configuration (DCI), read management (RM) and application level event standard (ALE) [9]. ALE is the sub-system that applies AC policies.

The Table 1 contains different implementations of the ALE sub-system, included on EPC global middleware, as it is established by the following specifications [10,11]. These traditional AC systems present two major challenges for supply chain application environments: (1) almost all include role-based access control (RBAC) as AC model and (2) the implementations are based on centralized architectures. Therefore, from the technical point of view, our proposal consists of an ABAC system for RFID systems that executes access control (AC) policies from a decentralized application (DApp) based on a blockchain architecture. Our proposal integrates several technologies, which allow in the first place the tracking of assets, i.e., an asset (e.g., SMI) is associated to a GTIN code. The system allows us to verify the existence of a certain asset based on the coding scheme presented. Finally, the proposal makes it possible to permit or deny access of an asset to a certain area (e.g., surgery room).

For this, smart contracts are used as an interface between the DApp and the blockchain, i.e., all these functionalities, including the AC policy, are executed from the DApp, which interacts with the smart contract, which, in turn, interacts directly with the blockchain, e.g., by a method to insert assets or a method to query certain attributes.

The remainder of the article includes the related work section, which is the keystone for the design of our system based on the reviewed literature, allowing us to arrive at conclusions. From this point onwards, we present the technical proposal, followed by the evaluation methodology, the results obtained and finally the conclusions and future research lines.

**Table 1.** Electronic product code (EPC) global standard: application level event standard (ALE) version and access control model [12].

| Middleware | ALE Version | Access Control Model |
| --- | --- | --- |
| IBM WebSphere RFID | 1.1 | Role-based access control (RBAC) |
| Oracle sensor edge server | 1.0 | RBAC |
| Rifidi Edge | 1.1 | RBAC |
| Fosstrak | 1.1 | - |
| Chuck | 1.1 | RBAC |
| Aspire | 1.1 | Access Control API[1] |
| WinRFID | 1.1 | - |

[1] API: Application Programming Interface.

## 2. Related Work

In this section, we present the literature justification that allows us to establish the use of an access control model based on attributes (ABAC) over an access control model based on roles (RBAC) for our use case. In addition, we indicate the preference of decentralized architectures over centralized architectures for supply chain environments and our use case. Additionally, we justify the blockchain selection within the set of distributed ledger technologies (DLT). Next, the type of blockchain that best fits our proposal is analyzed. Once the blockchain has been selected, if this is a public blockchain, a business model must be associated with it for implementation to be feasible. For this reason, we present a proposal based on an asset tokenization model for healthcare environments. However, the implementation of the tokenization model is established for future research lines. The related work section concludes with a discussion sub-section.

### 2.1. Attribute-Based Access Control (ABAC) vs. Role-Based Access Control (RBAC)

Although the RBAC model is well established, Gartner predicts that by 2020, 70% of companies will use ABAC to protect critical assets [13]. In addition, the following references [14–16], provide some clear limitations for the RBAC model such as:

(1) It is not possible to configure rules using parameters, which are unknown to the system.

(2) Permissions can only be assigned to user roles, not to objects and operations.

(3) Since the RBAC model is predominantly based on static organizational positions, there are problems in particular RBAC architectures where dynamic AC decisions are needed.

(4) It is possible to restrict access to specific system actions but not to data model.

(5) RBAC does not support multi-factor decisions (e.g., decisions that depend on location and timestamp).

On the other hand, the ABAC model presents important benefits that are adapted to our use case (Section 2) such as:

(1) ABAC provides access based on the attributes of each system component and not based on the user function [14].

(2) ABAC supports AC decisions without previous understanding of the object by the subject or understanding of the subject by the object owner [15].

A comparison that includes other features can be analyzed in Table 2. As a conclusion, we can establish that the ABAC model is suitable for our case of use based on the supply chain, i.e., applications that require flexibility and scalability.

**Table 2.** Comparison between RBAC and ABAC [14].

| Characteristic | RBAC | ABAC |
|---|---|---|
| Flexibility | Yes (For small and medium-sized organizations) | Yes |
| Scalability | No | Yes |
| Simplicity | Easy to establish roles and permissions, hard to maintain the system for a big company | Hard to establish all the policies at the start, easy to maintain and support |
| Support simple rules | Yes | Yes |
| Support complex rules | Yes | Yes |
| Support rules with dynamic parameters | No | Yes |
| Customizing user permissions | No (Every customization requires creating a new role) | Yes |
| Granularity | Low | High |

*2.2. Decentralized Model vs. Centralized Model*

As it can be analyzed in reference [9], most middleware implementations are based on centralized architectures. In order to examine the disadvantages of this model we use a common application environment for RFID systems such as the supply chain. Although the centralized approach is well adopted, it is not scalable, introduces bottlenecks and makes difficult to synchronize information, e.g., product status among different parts with their centralized DBs or to add new elements [17]. In addition, this model does not provide the degree of trust that must exist between the parties and therefore someone who is accountable for the data shared [18].

Different works focus on attribute-based access control models on centralized architectures. For instance, the reference [19] presents an AC model for IoT, in which it is established a coupling between ABAC and trust concepts. In addition, the reference [20] promotes an ABAC mechanism, which is applied to give the system the ability to implement policies to detect any unauthorized entry.

On the other hand, a decentralized model provides a solution to the aforementioned problems: firstly, the supply chain adopts a method in products can be tracked through every step of the chain, from suppliers, through manufacturers, to end users and secondly, with a certain degree of trust between the parties. Although a model based on decentralized architecture is the solution, the type of architecture to be used must be studied in depth, above all, to establish selection criteria.

2.2.1. Blockchain over Other Distributed Ledger Technologies

Blockchain technology is now entering a maturity stage that determines the use cases where the technology is applicable, which determines even the type of blockchain to be used. However, blockchain is not the only type of DLT, e.g., directed acyclic graphs (DAG) is considered another way to represent the data structure with advantages over the blockchain approach [21]. Therefore, we want to emphasize below the reason why the blockchain is suitable as a decentralized solution. First, it is clear that all data are not located on a central server, but are decentralized. These are distributed across all devices connected to the blockchain, so the blockchain can be thought of as a network of

nodes from peer-to-peer where a device (e.g., miner device) connected to the blockchain as the node, which talks to all the other nodes. In addition, this device will share the same responsibilities as the other peers and it will get a copy of all the data that is shared across the blockchain. All of this data is contained in packets of records called blocks that are chained together to create a "public" ledger and all of the network nodes work together to ensure that all of the "public" ledger data remains secure and unchanged and this is important for a AC application. The blockchain is fundamentally a DB and because all nodes communicate with each other in the blockchain, it is a network, so instead of the traditional centralized model, it is possible to think on a blockchain as a network and a DB all in one [22]. Once it is determined that blockchain is the type of decentralized architecture to implement, we focus on defining the type of blockchain suitable for our use case.

### 2.2.2. Selecting the Blockchain Type

Although the technical criteria of selection is fundamental, we first review the existing proposals in the literature and then the technical selection criteria. In that sense, there are some proposals that use AC based on blockchain, including RBAC. For example, the reference [23], proposes an approach based on blockchain to publish policies expressing the right to access a resource and to allow the distributed transfer of that right between users. In addition, the reference [24] includes a dynamic access control scheme for direct data communication between Internet of Things (IoT) devices. The reference [25] presents a RBAC using Smart Contract to realize trans organizational utilization of roles. Finally, transaction-based access control (TBAC) is a platform that integrates the ABAC model and the blockchain, combining four types of transactions and Bitcoin-type cryptographic scripts to describe the TBAC access control procedure corresponding to subject registration, object holding and publication, access request and grant [26]. By analyzing existing proposals that combine access control models in decentralized architectures, we conclude that these are mostly based on RBAC models. In addition, the proposal found based on ABAC uses Bitcoin as a blockchain. The technical criteria are detailed below.

The selection of the type of blockchain depends on factors such as the use case, the technical requirements and even the business model. For this reason, firstly it is considered the most recent Gartner recommendations, which indicate that to ensure a successful blockchain project, it is necessary to focus on the business problem, not on the technology solution [27]. According to the use case and the characteristics of the technological project, it is necessary to select between: a model based on governance with some trust between the parties and a certain level of centralization, represented by the Hyperledger Fabric Blockchain (HFB) or a model where there is no trust between the parties and fully decentralized, represented by the Ethereum (ETH) blockchain. The Table 3 presents common criteria to establish a comparison among different blockchain types. From our use case of a supply chain based on healthcare environments and taking into account, the project scalability, community support, skill availability, multi-functionality and adaptability, we consider deploy our DApp on the ETH blockchain.

From the decision (type of blockchain) based on technical criteria and taking into account the literature review carried out, we can affirm that our proposal contains high novelty value. The following sub-section summarizes all the points analyzed throughout the related work section, and based on our selection criteria and the revised bibliography; we arrive at conclusions and introduce a model.

### 2.3. Discussion

From the analysis performed throughout this section, we determined that ABAC is a suitable access control model for applications requiring flexibility and scalability. In addition, we analyzed that our use case is not optimal to build a centralized application for two reasons. Since our system is based on the ABAC model, asset attributes can change at any time, so that decision support is highly scalable. In addition, all the code in the application could change at any time and this means that the rules in AC policy also could change. Additionally, blockchain has been selected as DTL, thanks to features

such as immutability, necessary to ensure both AC and a reliable history of asset attributes behavior. Based on common criteria of selection, we determined the type of blockchain suitable to our proposal: the ETH blockchain.

**Table 3.** Comparison between popular blockchain types and a centralized database [17].

| Description | Public | Permissioned | Private | Centralized DB |
|---|---|---|---|---|
| Participation | Anyone | Members of organizations | Members of organizations | Limited |
| Write permissions | Granted | Restricted | Restricted | Restricted |
| Read permissions | Granted | Granted | Restricted | Granted |
| Speed | Slow | Fast | Fast | Slow |
| Identity | Anonymous | Anonymous | Known | Known |
| Security | Impervious to security attacks | Impervious to security attacks | Impervious to security attacks | Vulnerable to security attacks |
| Transparency | Visible across all supply chain nodes | Visible across all supply chain nodes | Restricted to specific supply chain nodes | Restricted to specific supply chain nodes |
| Traceability | Yes | Yes | Restricted | No |

However, if we take up Gartner's recommendation [27] to be able to deploy our proposal on the main ETH network, even if the project is technically feasible, it needs to be endowed with a business model that makes it viable. For that reason, we present below one based on asset tokenization.

The tokenization through a blockchain platform (the most used is ETH) enables us to leave not only the use of expensive and complex transactions, but also the exchange itself. Any person enrolled in the blockchain could potentially act as an issuer of a legitimate asset that he or she would like to tokenize [28]. Applied to healthcare, tokenization can contribute to reducing the cost of private medical treatment by transferring the ability to maintain and hold data from intermediaries, like insurance companies, hospitals and pharmacies to patients. In the existing scheme, neither of these subjects share information with patients, and patients are unable to verify the data's correctness. Through tokens, both patients and the general public can keep their data and share it with anyone they want [29]. Tokenization can also automate the payment process. In addition, since tokens are a secure and protected way to make transactions, the payment system is simplified. However, the main challenge is that, so far, no nation has a strong regulation for cryptocurrency. As a result, tokens do not have legal rights to property and are not protected by law. Therefore, legislative changes are required to adapt these new business models [30].

Therefore, a business model based on tokenization is applicable to a public blockchain such as ETH, which it contextualized to healthcare through the supply chain and, therefore, to our model based on control and traceability. A tokenization model is applicable to both assets (e.g., SMI) and for patient information (e.g., blood pressure sensors). Thus, a security model based on AC is highly suitable and applicable in these environments.

Since our article is a proposal, the technical behavior of our implementation evaluates firstly in a local environment (i.e., our own ETH node, without joining the main ETH network) and secondly scales to an ETH Testnet. Ropsten Ethereum, also known as "Ethereum Testnet", is a testing network that runs the same protocol as Ethereum and is used for testing purposes before deploying on the main network (Mainnet). In order to scale our proposal to the Mainnet, we will propose a tokenization-based model, which we introduced above and is part of our future research lines.

### 3. Proposal

As we mentioned in the introduction, our proposal consists of a complete system in which several technologies converge. However, we want to start this section with a basic architecture that enables a general understanding of how our system performs ABAC.

### 3.1. Decentralized System Architecture

Figure 3 represents the general architecture of the proposed ABAC model based on ETH blockchain. The physical node is composed of the RFID Reader Control (RFID-RC), the DApp and the smart contract. When a medical instrument (previously tagged with an RFID tag) attempts to gain access to a room, the RFID-RC sends a request for access to the DApp. The DApp sends a query via smart contract to the blockchain network, which returns some attributes related with the asset (e.g., company prefix, product type, serial number). In addition, the DApp receives other attributes (e.g., timestamp) from the RFID-RC. Then, the DApp uses the attributes to execute the ABAC security policy, which determines whether tag access is permitted or denied. The next section details the implementation framework used. Additionally, we need to confirm that one of the main advantages of a decentralized system is scalability, so this physical node (smart oracle) can be replicated in a way that establishes a new connection with the blockchain (via smart contract), without affecting any of the existing nodes.

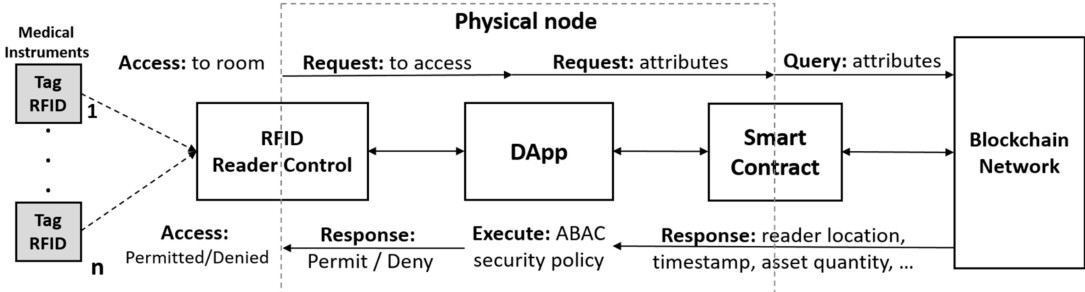

**Figure 3.** Decentralized system architecture based on Ethereum (ETH) blockchain.

### 3.2. Access Control Mechanism (ACM)

For a subject to be able to execute a policy on an object (e.g., permit or deny access), ABAC access control mechanism (ACM) must be enabled. ACM includes the next steps: (1) check the subject's attributes, (2) check the AC policies (rules), (3) check the object's attributes, and 4) check the environmental conditions. Although it is normal to expect that the subject is a human, a non-person entity (NPE), such as an autonomous service or an application, could also occupy the subject's role, as the reference [15] indicates. In our case, the reader requests the DApp for the tag RFID (associated with a SMI) access.

Before analyzing the AC policy, some boundary conditions are established for the transfer of an asset from the source room (e.g., sterilization area) to the destination room (e.g., surgery room) and vice versa should be mentioned:

(1) The transaction that authorizes the transfer of an asset is invoked by an authorized employee through a trusted application connected to DApp (Figure 4).

(2) The tag uses an EPC code with a pattern similar to the Expression (1) and illustrated in Expression (2):

$$01.0000389.000162.000169740$$
$$\text{Header|CompPrefix|Product type|Serial Number,}$$

(2)

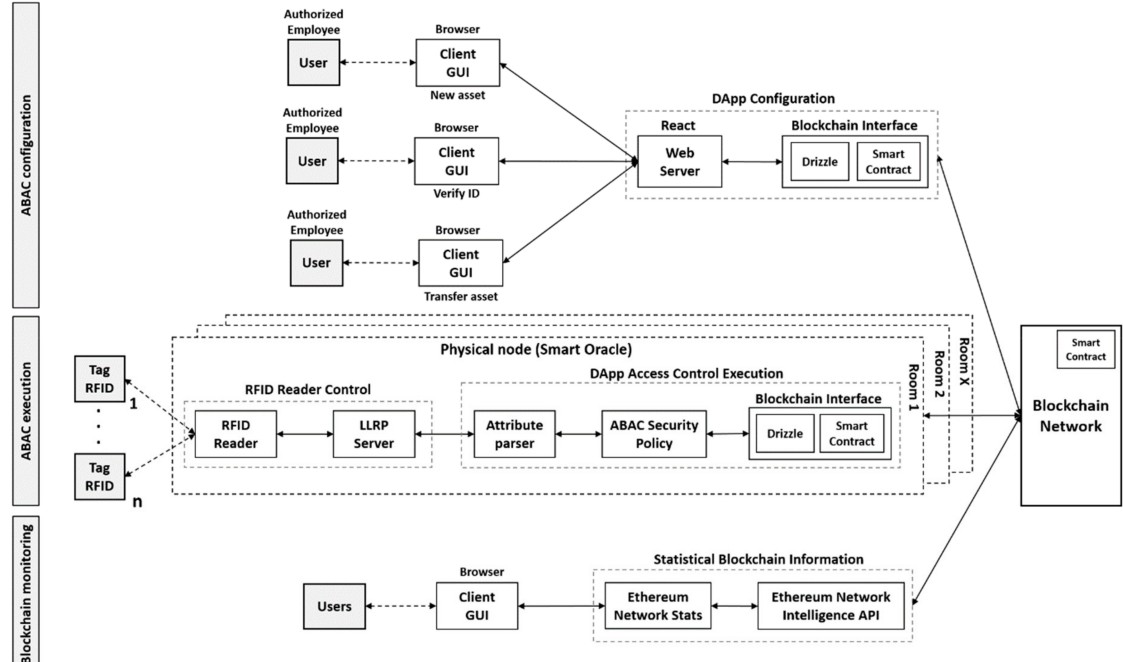

**Figure 4.** Details of the system architecture.

The process that is performed by DApp when it receives an access request is described next. The variables' names used to define the AC policy are included. (1) The subject (reader) is verified based on two attributes: reader name (variable 01: "rdr_nm", e.g., rdr_nm: "roomA") and location (variable 02: "loc", e.g., loc: "41.40338, 2.17403")). (2) The company prefix (variable 03: "cmp_prf", e.g., cmp_prf: 000389), the product type (variable 04: "item_ref", e.g., item_ref: 000162), the serial number of a specific asset (variable 05: "ser_nmb", e.g., ser_nmb: 000169740) and the asset status (variable 06: "st", e.g., st: "STERILIZED") are verified. (3) The environmental condition is verified based on the time elapsed since an asset is sent to an existing reader in a medical room (through a transaction) and that reader receives the request for access to that asset (tag). The environmental condition is approved if the interval is less than 10 min (600 s). This time is set for moving assets between locations once the transaction has been invoked. In that sense, variable 07: "time_in" (e.g., time_in: 1,560,209,335) is the time record once the transaction is completed and variable 08: "time_out" (e.g., time_out: 1560209455) is the time given when the reader requests access to this RFID tag.

Based on the AC policy notation established in the reference [31], our AC policy C is defined in the Expression (3).

We decided to implement the AC policy on the DApp and not as part of the smart contract for two reasons. Firstly, as we indicated in Table 3, one of the constraints of a public blockchain is the speed, so if the AC policy is executed as part of the smart contract, it would lead to a delay. Secondly, since smart contracts are public the AC policy would be exposed. In this way, one of the future research lines is the implementation of this model in a private blockchain (e.g., HFB); so that the AC policy can be located within the smart contract (chaincode) in order to analyze these results.

$$C = \begin{cases} Ture, \quad if : (\text{rdr}_{nm} = \text{"roomA"} \cap \text{loc} = \text{"41.40338, 2.17403."} \cap \\ \text{cmp\_pfr} = 000389 \cap \text{iem\_ref} = 000162 \cap \text{ser\_nmb} = 000169740 \cap \\ \text{st} = \text{"STERILIZED"} \cap \text{time\_out} - \text{time\_in} \leq 600) \\ \qquad\qquad False, \qquad otherwise \end{cases} \quad (3)$$

### 3.3. Technical Implementation Details

The two previous sub-sections allowed respectively to define the general functioning of our ABAC model and to detail how the AC policy is executed in the DApp; therefore, it is time to present our system in detail. In order to better understand it, it begins with a summary of the main technologies used.

Table 4 summarizes the technologies used in each sub-systems and their associated blocks. Table 4 follows the Figure 4 design principles. For a better understanding of our work, the reference [32] is a film reference included as external document.

In order to analyze the technical implementation details, Figure 4 shows the specific architecture of our system, which consists of three sub-systems: ABAC configuration, ABAC execution and ETH blockchain monitoring.

**Table 4.** Technologies used.

| Sub-System | Block | Implementation | Technology/Library |
|:---:|:---:|:---:|:---:|
| ABAC Configuration | New asset | X | ReactJS |
| | Verify ID | X | ReactJS |
| | Transfer asset | X | ReactJS |
| | Blockchain Interface | X | Truffle-Drizzle |
| ABAC Execution | Rifidi Virtual Reader (http://wiki.rifidi.net/) | [33] | Java |
| | LLRP Server | X | llrp-nodejs [34] |
| | Attribute Parser | X | node-epc [33] |
| | ABAC Security Policy | X | Java Script |
| | Blockchain Interface | X | Truffle-Drizzle |
| Blockchain monitoring | ETH Network Stats | [35] | AngularJS |
| | ETH Network Intelligence API | [36] | Java Script |
| Blockchain local node | Smart Contract | X | Solidity |
| | ETH Network | X | Geth |
| Legend | "X" implies that we implements the block based on the technology or library | | |
| | "[ ]" implies that we use the project. | | |

### 3.3.1. ABAC Configuration

The ABAC configuration sub-system includes a graphical interface (GUI), based on ReactJS (https://reactjs.org/) web technology that is launched from a browser (Figure 5). This GUI includes two views (Figure 5). Since a demonstration environment is presented, the two views are included within the same browser window, however as it is detailed, each view has its own functionality. The first view allows an authorized employee to add new assets to the system. This employee introduces the code of the company prefix, the code of the product type, the asset ID (e.g., serial number) and so on (Figure 5). Each time a new asset is stored in the ETH blockchain, a new transaction is generated. In order to transfer an asset between rooms the authorized employee first needs to verify the ID (e.g., serial number) of the asset, through a simple query to the blockchain via smart contract. To do this, the authorized employee uses the button ("VERIFY ID") of the second view. This blockchain query does not generates transactions. Next, the same second view enables the transfer assets from the source room (e.g., sterilization area) to the destination room (e.g., surgery room), before attributes values, such as asset status (e.g., "STERILIZED") and timestamp are updated (Figure 5). This action is carried out from "TRANSFER ASSET" button. Since asset transfer involves changes (e.g., room, status, timestamp) new transactions are generated via smart contract. Details of blockchain interface operation are analyzed in the following sub-section.

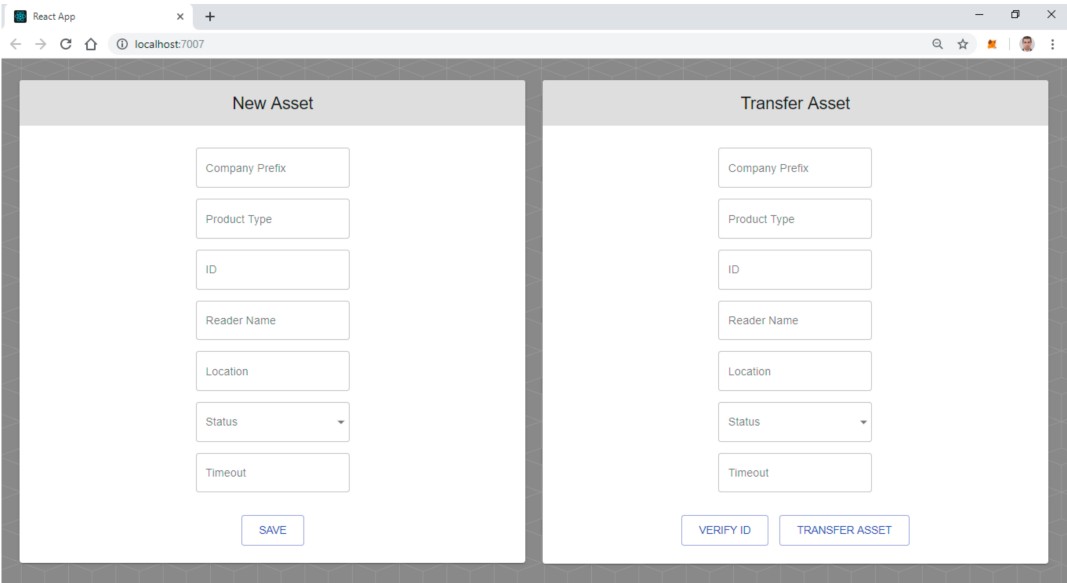

**Figure 5.** Graphical interface (GUI) of ABAC configuration sub-system.

### 3.3.2. ABAC Execution

The ABAC execution sub-system contains a smart oracle to permit or deny asset access and it is located in each of the medical rooms. Our smart oracle includes the RFID reader, the LLRP server, the attribute parser (AP), the ABAC security policy (ABAC-SP) and the blockchain interface (BI). The AP, the ABAC-SP and the BI comprise the DApp access control execution (Figure 4, ABAC execution sub-system) and RFID-RC includes the RFID reader and the LLRP server (Figure 4, ABAC execution sub-system).

The RFID reader interacts directly with the tagged assets and the LLRP server. LLRP is a protocol that EPC global ratified as a standard that constitutes an interface between the reader and its software or control hardware [37]. The protocol sends XML (eXtensible Markup Language) messages between the client (e.g., RFID reader) and the server (e.g., LLRP server). To develop our proof of concept (PoC) we use an open source tool, as known as Rifidi ([38]), that create a virtual reader and RFID tags based on SGTIN96 standard. Details of the project that supports it, as well as the getting started guide are located in [33]. In addition, since our LLRP server is based on the standard LLRP, it is agnostic to any RFID reader that supports the LLRP protocol such as Motorola FX7400, Intermec IF61and Impinj Speedway.

The AP receives the RFID Tag EPC from the LLRP server and uses a GTIN conversion system, based on a NodeJS library [34], which allows transforming the RFID TAG EPC code to the EPC Tag URI (Uniform Resource Identifier) (e.g., Expression (4)). AP filters the attributes: Company Prefix (variable "cmp_prf" in AC policy), Product Type (variable "item_ref" in AC policy) and Serial Number (variable "ser_nmb" in AC policy). In addition, AP controls other attributes such as timestamp (variable "time_out" in AC policy), reader name (variable "rdr_nm" in AC policy) and location (variable "loc" in AC policy).

$$\text{RFID Tag EPC: 3074257bf7194e4000001a85} \tag{4}$$
$$\text{EPC Tag URI: urn:epc:tag:sgtin-96:3.0614141.812345.6789,}$$

The BI is built based on the truffle framework, using the drizzle library to interact with the web3.js server. Drizzle is a collection of front-end libraries that enable writing DApp front-end in an easier way [39]. The communication is performed between the parties via GET and POST methods. For instance, ABAC-SP determines whether asset access is permitted or denied, it sets a variable, which

is sent via POST method to the LLRP server. Therefore, the LLRP server sends an XML "keepAlive" message (Figure 6) to maintain the interaction with the RFID tag or simply disconnects it.

To execute the AC policy established by the Expression (3), ABAC-SP matches the attributes from the AP with the attributes queried from the blockchain.

```xml
<?xml version="1.0" encoding="UTF-8"?>
<llrp:SET_READER_CONFIG_RESPONSE xmlns:llrp="http://www.llrp.org/ltk/
  <llrp:LLRPStatus>
    <llrp:StatusCode>keppAlive</llrp:StatusCode>
  </llrp:LLRPStatus>
</llrp:SET_READER_CONFIG_RESPONSE>
```

**Figure 6.** (eXtensible Markup Language) XML keepAlive messages to permit the access.

### 3.3.3. Ethereum (ETH) Blockchain Monitoring

Although this sub-system is an integral part of our implementation ([32]), how the following section is dedicated i.e., it enumerates and describes the monitoring tools of our system in order to verify its feasibility, we have preferred to set the analysis of this sub-system as part of the successive sections. In that sense, Figure 7 represents the monitoring tool ETH Network Stats, which as part of this sub-system.

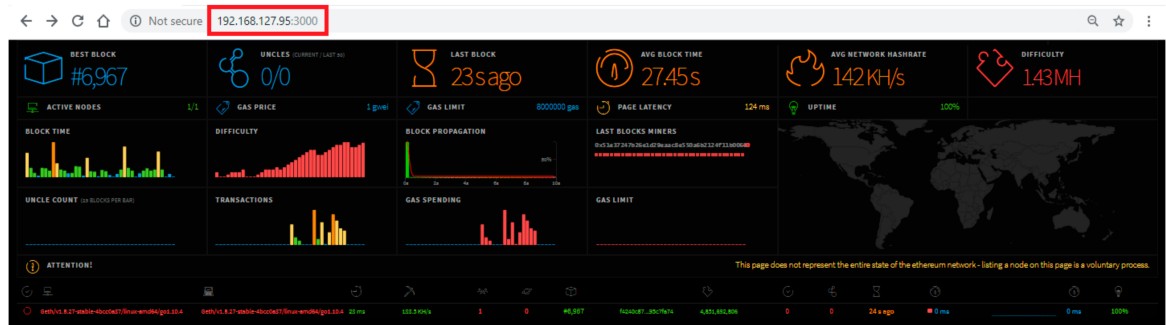

**Figure 7.** ETH Developer tools List.

## 4. Evaluation Methodology

In order to evaluate the feasibility of our proposal, it is necessary to indicate first that our model has been deployed in two environments, one based on local blockchain and the other based on a Testnet blockchain.

In the first case, an ETH node was deployed, although it included the property of no discover, making it impossible to connect to the Mainnet. The expression (5) is a sample of the command deployed based on geth (https://geth.ethereum.org/) client (main ETH client).

$$geth\ –datadir\ data\ –unlock\ 0x8a6d63ea98e05a550b01f8aa4a19021e43bd43f0\ –networkid\ 123456$$
$$—ws\ -wsaddr\ 192.168.127.95\ –wsport\ 8546\ –wsorigins\ "*"\ -rpc\ -rpcaddr\ 192.168.127.95 \quad (5)$$
$$–rpcport\ 8545\ –rpccorsdomain\ "*"\ –nodiscover\ console\ 2>>\ ETH.log,$$

In the second case, to scale our system, as mentioned above, we use Ropsten as Testnet. Some of the advantages that have allowed us to select this network over others like Kovan, Rinkeby and Sokol are:

(1) It better reproduces the current production environment, i.e., the system and network conditions in the Mainnet, since it employs the proof of work (PoW) as the consensus algorithm between the nodes.

(2) It can be used with both geth client and parity client.

(3) It enables to join its own node to the network, i.e., to participate in the PoW or simply request the ether from a faucet (https://faucet.metamask.io).

To access the network it is necessary to create an Infura project, which generates the endpoint URL (for example, the expression (6)) used in the configuration files of our system (truffle-config.js). Next is a detail of the tools used and then the features that are measured:

$$\text{ropsten.infura.io/v3/fa42299dbea54014801bc4145d7a1a1e,} \qquad (6)$$

*4.1. Evaluation Tools*

First, we present the tools used to evaluate our system: ETH Network Stats, Etherscan, Truffle Test, and Infura Dashboard. These tools have been deployed both for the local environment and for the Testnet. The exception is Infura which is a tool that is only associated with the Testnet. Below is a brief description of the tools.

ETH Network Stats is a tool composed by a front-end Ethereum Network Stats [35] and a back-end Ethereum Network Intelligence API [36]. This is a visual interface for tracking Ethereum network status. It uses WebSockets to receive stats from running nodes and output them through an angular interface. Both servers are installed locally. This tool was presented as part of the sub-system ETH blockchain monitoring. ETH Network Status ([40]) is the equivalent to ETH Network Stats, but used to the Ropsten Testnet.

Etherscan Ropsten Testnet Network ([41]) is a tool that we will use to monitor the state of the blockchain and the transactions that are stored in it. This tool presents an equivalent for the local environment, which it is installed as a server.

Infura Dashboard is a response to developer demand for a better understanding of how to improve DApps. The following reference [42] mentions that it has been recently updated, enabling us to obtain relevant information about calls to web3.js methods, which allow for some type of interaction (e.g., generating a transaction) with Ropsten Testnet.

Truffle test is combined with the data obtained from contract migration process in order to improve the data analysis. Truffle comes standard with an automated testing framework to make testing the smart contracts easy. This framework lets it write simple and manageable tests in JavaScript or Solidity. The JavaScript way is used from the outside world, just like an application. The Solidity way is used in bare-to-the-metal scenarios. Truffle test has been deployed for both the local environment and the Ropsten Testnet.

The Table 5 summarizes these tools with application environment and the main features measured.

**Table 5.** Tools used to test the proposal.

| Tool | ETH Network Stats/ETH Network Status | Etherscan | Truffle Test | Infura Dashboard |
|---|---|---|---|---|
| Features | Network monitoring | Blockchain monitoring | Smart Contract monitoring | Bandwidth monitoring |
| Local Environment | Network monitoring of our local node (Figure 7) | Local ETH Blockchain monitoring (e.g., contract addresses, transactions, blocks) | Testing the smart contract interaction with local blockchain | - |
| Ropsten Testnet | Network monitoring of Ropsten Testnet | Testnet blockchain monitoring (e.g., contract addresses, transactions, blocks) | Testing the smart contract interaction with Testnet blockchain | It allows seeing the bandwidth behavior for each web3.js method used |

*4.2. Measurements*

We consider the analysis as integral because can test each part of the implementation, i.e., from network monitoring, with features like the number of nodes and the network hashrate to the delay of

the smart contract application and the bandwidth consumption for each web3.js methods. Below we present the main parameters that can be monitored through the tools listed in the previous sub-section. Considering the main characteristics associated with each tool, Figure 8 establishes the logical order of use of these tools with respect to the features analyzed.

ETH Network Stats and ETH Network Status allow measurement of a wide range of parameters within the ETH network. These parameters focus mostly on network status (Table 5). Some of the parameters that can be measured are number of successfully mined blocks, presence of uncle blocks, mined time of the last mined block, average mined time, average network hash-rate, difficulty, active nodes, gas price, gas limit, page latency, uptime, node name, node type, node latency, peers connected to one node and some others. Figure 7 illustrates the tool in use, accessed from the browser from a private IP on port 3000.

As we mentioned Etherscan allows extraction of information relative to the blockchain (Table 5). Within the parameters that can be obtained are account balance, account info, transaction hash, block number, type of token (e.g., Erc20), average gas used, transaction costs and transaction fee.

Infura Dashboard allows obtaining a wide range of parameters such as: the total number of methods called, the bandwidth consumed by each of the methods used and the total bandwidth consumed. Therefore, the main feature measured is the bandwidth (Table 5).

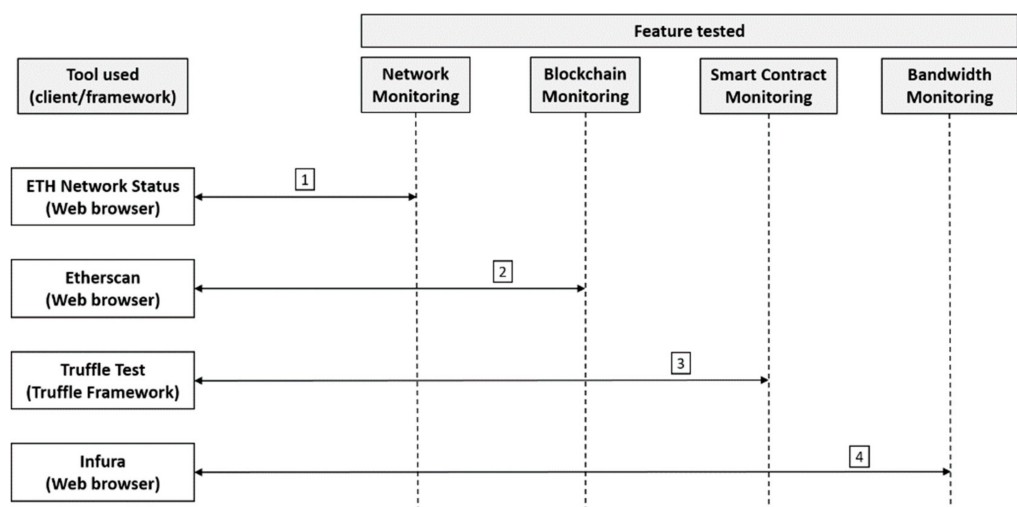

**Figure 8.** Sequence diagram of tested features and used tools.

As mentioned, the truffle test is a framework that allows running tests on smart contracts. For our case of use, the parameters we measure are time of data query, time and cost of data insertion and time of full test. In addition, these data are combined with the data obtained from contract migration process. Therefore, other parameters measures are gas and time spend to deploy the contract. The results obtained are presented below.

## 5. Results

Based on the analysis performed in the previous section, the results are presented for each of the tools.

ETH Network State and ETH Network Status enables monitoring the network all the time. For example, at the time of analysis the Testnet Ropsten has mined 5,931,224 blocks, has 14 active nodes, the average block time is 14.04 s, the average network hashrate is 120.1 MH/s, and the difficulty is 2.16 GH. These parameters can be contrasted with those shown in Figure 7 for our locally deployed blockchain. For instance, at the time of analysis our local blockchain has mined 6967 blocks, has only one active node (our node), the average block time is 27.45 s, the average network hashrate is 142 KH/s and the difficulty is 1.43 MH. As a conclusion, it is visible that the power of mining and therefore the

resources available to our local device are much less than those presented by the public network. This is an expected result.

Etherscan Ropsten Testnet Network ([41]), which allows us to have a view of all transactions that have been executed from our test address (e.g., 0xe8d5487caebfb3f3e93304161cad0d5d3078b033). Other attributes that can be verified are the status of each of the transactions, the block where the transaction has been assigned, the gas percentage used (e.g., average gas used 66.67% of the established limit value), transaction costs and fee, as well as the nonce used in the PoW. Similar behaviors are obtained for the tool used locally.

Figure 9 is taken from the Infura dashboard and it details the main methods called by the web3.js library in order to interact with the blockchain via Smart Contract, as well as the bandwidth they spend. Clearly, there is a relationship between the method that infura detects and the method we use in our blockchain interface (BI) based on the truffle-drizzle framework, only that Infura perceives JSON (JavaScript Object Notation) RPC (Remote Procedure Call) API (https://github.com/ethereum/wiki/wiki/JSON-RPC) methods based on web3.js library and, since truffle-drizzle works with promises and web3.js works with callback, truffle-drizzle framework uses functions like *cacheSend*. Calling the *cacheSend* function on a contract will send the desired transaction and return a corresponding transaction hash so the status can be retrieved from the store. The procedure mentioned at web3.js level is performed by *eth_getTransactionByHash*, however since we work at a higher level, our *cacheSend* function agglutinates this and other methods. On the other hand, when performing debugging via node inspect tool on the migration process (truffle migrate), it is determined that both methods: *getTransactionReceipt* and *eth_getCode* are used. This highlights the importance of the Infura dashboard and details the consumptions made by the methods at a low level. In addition, Infura dashboard includes other relevant information such as peak (e.g., 183.33 MB) and average (e.g., 9.11 MB) hourly bandwidth usage and so on.

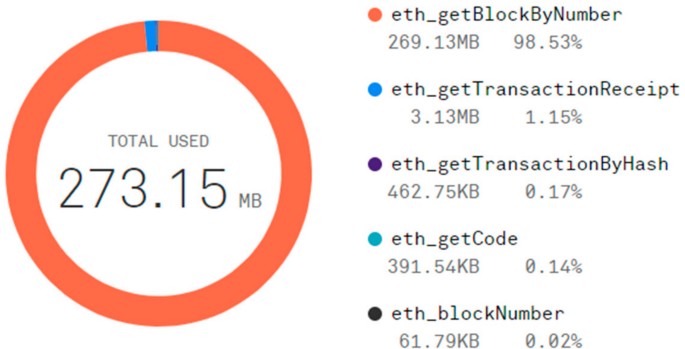

**Figure 9.** Infura dashboard tool: top five methods call bandwidth usage.

In order to examine our smart contract, we have established a combination of tests between the truffle test frame and the data obtained from the contract migration process. Therefore, Figure 10 shows the succession of the applied mechanism in order to check the feasibility. The data received are shown in Table 6, which compares data insertion time, data query, time gas used and time to pass the full test. Expression (7) is used to calculate the percent. Since, the times achieved are not deterministic, was taken both best and worst times. The test process performed is described below.

$$(Local\_network\_time/Ropsten\_network\_time) \times 100, \tag{7}$$

Our model initiates by recovering the migration time from the contract, which is not deterministic and it establishes a considerable delay between migrations in a local environment and migration in the Ropsten Testnet. Although it is a time to consider, it is not decisive to evaluate the feasibility of the system, since its implementation is prior to the deployment of the system. Because the same

smart contract is employed in both environments, the costs and computational power required for the deployment match.

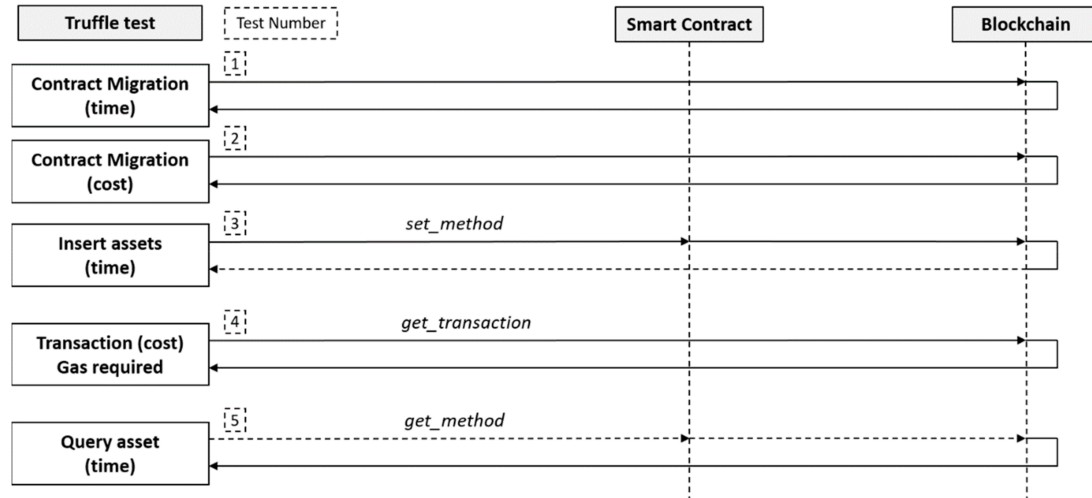

**Figure 10.** Sequence diagram of truffle test and contract migration.

Another essential requirement is the insertion time of the asset attributes. The *set_method* is used to send attributes to the deployed smart contract and it waits for blockchain response. This procedure is equivalent to the mentioned one for inserting data via GUI (ABAC configuration sub-system, Figure 5). Although the delay between the local environment and the Ropsten Testnet is evident (Table 6), this metric will not cause a delay in the execution of the AC policy. As mentioned above, data insertion involves the generation of transactions and, therefore, associated costs, which is an indispensable measurement, so the get_transaction method is applied. We consider that since the same smart contracts deployed in different environments, the transaction cost is equivalent.

**Table 6.** Truffle test results, local network vs. Ropsten network.

| Monitored Feature | Local | Ropsten | Local vs Ropsten (%) |
|---|---|---|---|
| Gas used to deploy | 732,151 | 732,151 | - |
| Cost to deploy the contract | 0.01464302 ETH | 0.01464302 ETH | - |
| Total migration cost | 0.01987088 ETH | 0.01987088 ETH | - |
| Contract Migration time | 15,703 ms | 225,001 ms | 6.98 % |
| Data query (best time) | 112 ms | 855 ms | 13.1% |
| Data query (worst time) | 160 ms | 1,705 ms | 9.38% |
| Data insertion (best time) | 184 ms | 12,225 ms | 1.505% |
| Data insertion (worst time) | 205 ms | 40,646 ms | 0.46% |
| Gas used to data insertion | 43,255 | 43,255 | - |
| Time passing full test (best time) | 814 ms | 54,000 ms | 1.507% |
| Time passing full test (worst time) | 902 ms | 180, 000 ms | 0.501% |

The decisive metric (delay requirement over AC policy) is the query of assets data. Therefore, it is essential to execute the get_method function and wait for the delay value (Table 6). For this reason, we can conclude that the implementation of our system is technically feasible. Since the Ropsten Testnet is less stable than the Mainnet because of the smaller number of nodes joining that network, and, therefore, less computational power, as discussed in tool one (ETH Network Status), these delay would be less in a Mainnet.

## 6. Conclusions

The growing adoption of RFID systems in healthcare is evident. Based on interviews with specialists we determine the implementations needs of a trust tracking and tracing system of medical assets. Our proposal is an access control system of a healthcare asset in order to prevent unwanted assets from entering the wrong area because of human error or an external threat. Therefore, it is a prevention system that aims to solve both security and safety risks. Traditional access control systems are based on role-based access control (RBAC) and centralized architecture. From the technical point of view, our proposal consists of an attribute-based access control (ABAC) system for RFID systems that executes access control (AC) policies from a decentralized application (DApp) based on a blockchain architecture. This model is a proof of concept in both a local environment (single node) and in a public environment (Ropsten Testnet) and although the technological feasibility for its eventual production implementation is demonstrated, it requires a viable underlying business model. In order to demonstrate the implementation feasibility were used four recommended tools: ETH Network Status, Etherscan Ropsten Testnet Network, Infura dashboard and truffle test.

Future research lines are firstly, to establish a comparison between systems based on Hyperledger Fabric Blockchain and other with Ethereum blockchain. One of the common criteria in order to establish a comparison is the ABAC policy as part of the contract (Chaincode and Smart Contract). Secondly, to consider an application environment based on the public blockchain with a base on a tokenization environment. Thirdly, the creation of a robust mutual authentication RFID protocol that works together with our ABAC blockchain system in order to build a secure supply chain system. Finally, to extend ABAC and RBAC blockchain concept to industrial manufacturing and automation environments. Recently, modbus.org has established security requirements, which include RBAC authentication based on X.509v3 certificates.

**Author Contributions:** Investigation, S.F., J.A. and S.A.; Methodology, J.A. and S.A.; Software, S.F.; Supervision, J.A. and S.A.; Validation, J.A. and S.A.; Writing—original draft, S.F.; Writing—review & editing, J.A. and S.A.

**Funding:** This research received no external funding.

**Conflicts of Interest:** The authors declare no conflict of interest.

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
