# Peer review of "An Attribute-Based Access Control Model in RFID Systems Based on Blockchain Decentralized Applications for Healthcare Environments"

_computers, doi:10.3390/computers8030057_

Round 1
Reviewer 1 Report
The authors present a security proposal, working with an attribute-based access control using smart contracts in RFID systems. Below I am presenting my comments.
1/ The problem must be enhanced in the Abstract. Also, in the Abstract present, the contribution of the article clearly.
2/ In the Introduction, it is very important to present related work briefly. I am referring to ARTICLES, not technologies.
3/ Are you using Blockchain. Why not presenting this in the title? Regarding the title, the title is not good. It does not present your work in a clear way.
Four/ Present in a clear way the contributions in the Introduction.
5/ I did not understand Section 3. Why are you presenting it? The idea is to discussion possibilities or to present your model?
7/ In Figure 1, what your boxes and what are not? Detach them in colors. In other words, what already exists and what is not?
8/ Hot about system scalability?
9/ What are the limitations of your system?
10/ Are your model portable with other existing RFID solutions?
11/ In the Conclusion, highlight the contributions for the society. In other words, present the impact, the novelties, for the society (end-user).
12/ The References are basically web sites. Hot about articles? Remember, you are presenting an article for a scientific journal.
Author Response
Dear reviewer,
Thank you for the review and your comments. We consider that thanks to the comments, this manuscript has improved in content and value for the reader. We hope that the new information, references, demonstration, figures, tables and analysis will meet your expectations.

Reviewer 2 Report
The authors presented a Paper related to Attribute-based Access Control System using Blockchain for RFID systems.
The idea and the topic are absolutely interesting, the presentation quality is good and the application context has been introduced very well. A Decentralized approach combined together with Blockchain technology is appealing and challenging. However there are some issues that must be addressed before publication:
- I would suggest to add a Related Work Section in order to properly organize the existing works and paper in the field
- It is really important to better underline the proposed contribution compared to the state of the art
- An High Level figure may be really useful in section 1 to allow the reader to easily understand the context and the involved actors that commonly characterize an RFID platform. The same thing may be also useful at the beginning of Section 2 in order to introduce the Healthcare system.
- According to me Section 5 is the main issue of this paper. It is absolutely a weak section without many fundamental information. The implementation it is not clear in particular for the Blockchain part and Physical Note Section. Furthermore, no measurements about system performance are available and it is really hard to evaluate the quality of the proposed approach. What is the computational cost for the distributed architecture ? Spent time for each operation ? Delay and execution time ? How blockchain nodes are connected ?
Author Response

(The authors gave the same response as above.)

Round 2
Reviewer 1 Report
The Abstract was completely reorganized. Now, we can see the problem in the literature and the proposal of the article.
What is DApp in the Title? Fix it, right?
The organization of the article is strange. This because I expect: Introduction, Related Work, Proposal, Evaluation Methodology, Results and Conclusion. Why not presenting this?
Did you see the middleware named Eliot, from South Korea?
In the related work, highlight pain the gap in the literature.
What is the idea of Section 2? Is it your contribution?
Figure 2 is strange. Is it generic? Is it an example? How about passive or active RFID tags and readers?
You are presenting a proposal or a survey. It is not clear. Section 3 preens a lot of information.
I did not understand Section 3. Again, the ideas are not in the chain. Present your motivation in the introduction and in the related work sections.
In Figure 3, what are your contributions and what is from others? The same for Figure 4.
How about system scalability?
The conclusion was improved. The first paragraph is too long. Again, what is the contributions for the society, end-user? How did you achieve your scientific contributions?
Author Response
Dear reviewer, thank you for your comments, we attach the cover letter.

Reviewer 2 Report
The authors improved the paper after the first submission. However there are significant points/issues that should be addressed in the experimental analysis:
- The linked video is interesting and shows the implementation but from the reviewer and reader point of view a paper should be self consistent without the need to access an external resource in order to understand the contribution and the implementation. I would suggest to attach the video as external document in the submission and citation in the paper (with the Last Access information and not directly in the text) but bring some contents such as the UI directly into the paper in order to better explain the experiment.
- One or more Sequence diagrams may really help the reader to understand the involved steps for each phase of the evaluation/experiments and how the involved actors communicate and exchange information (this kind of information are interesting also together with the architecture discussion)
Author Response

(The authors gave the same response as above.)

Round 3
Reviewer 1 Report
The article was improved, both in term of structure and content. Now the ideas are clearer for me. I can suppose that the reader of the Journal will not have problems reading the article.
Reviewer 2 Report
I have read the last manuscript update after the additional review round.
I believe that the paper has been significantly improved in many different aspects.
According to me the paper can be accepted in the current form.